# Historic Graffiti as a Visual Medium for the Sustainable Development of the Underground Built Heritage

Mia Gaia Trentin [1,*], Doron Altaratz [2,*], Moshe Caine [2], Amit Re'em [3], Andrea Tinazzo [4] and Svetlana Gasanova [1]

1   APACLabs, STARC, The Cyprus Institute, Aglantzia 2121, Cyprus; s.gasanova@cyi.ac.cy
2   Department of Photographic Communications, Hadassah Academic College, Jerusalem 9101001, Israel; mosheca@hac.ac.il
3   Israeli Antiquities Authorities (IAA), Jerusalem 9100402, Israel; reem@israntique.org.il
4   Independent Researcher, Cumiana, 10040 Torino, Italy; andrea.tinazzo1991@gmail.com
*   Correspondence: m.trentin@cyi.ac.cy (M.G.T.); doronal@edu.hac.ac.il (D.A.)

**Abstract:** Since prehistoric times, graffiti has been a way for humans to express themselves and interact with the landscape in a visual way. Graffiti is a visual record of the relationship between society, culture, and the environment over time, representing an additional layer of sociocultural value to the underground built heritage (UBH). Thanks to the application of digital technologies and a specific workflow, this paper will suggest how graffiti can be regarded as an additional and relevant element of creating connections and strengthening the site's values, bridging the past and present communities. Through the critical discussion of two case studies—the monastery of Ayia Napa (Cyprus) and the Saint Helena chapel in the Holy Sepulchre in Jerusalem—the authors want to achieve two main goals: first, they want to highlight the sociocultural value and raise awareness about the presence and significance of historic graffiti. In the second instance, they wish to illustrate how graffiti can be an additional agent for the sustainable development, valorization, and promotion of the UBH.

**Keywords:** Eastern Mediterranean graffiti; graffiti in UBH; CH sustainability; imaging techniques for CH; visual communication; CH memory; sense of place



## 1. Introduction

Graffiti holds an essential place as a marker in humans' never-ending relationship with the landscape. For a long time, humans interacted with their surrounding space, marking it, leaving signs, and recording stories and events by scratching or painting any available and functional surfaces. From a diachronic perspective, this phenomenon presents and characterizes human history and evolution with different features and functions [1]. If, on one side, a general definition of graffiti is able to embrace such a long-lasting and varied phenomenon, it is difficult to elaborate [2], and on the other, common characteristics can be identified.

Graffiti fulfills humans' inner need to communicate, leave a sign, and spontaneously interact with their surrounding space without strict rules to follow concerning the graphic form, the content, or the location. From rock art to street art, graffiti constitutes an on-site graphic archive displayed on surfaces not intended for writing [3] (p. 12) [4] (p. 9). Graffiti, in this sense, creates an invisible layer covering the natural and anthropic landscape, recording and testifying to the human presence, interaction, and perception of those places. This strong connection and the founding relationship between graffiti and their context-space have been extensively highlighted [5] and can be defined by the concept of *graffitiscape*. The neologism, obtained by combining the two words *graffiti* and *landscape* in its cultural meaning, as defined by UNESCO (https://uis.unesco.org/en/glossary, accessed on 23 July

2023), indicates the *graphic* manifestation of the interaction between humankind and their natural and anthropic environment.

Moreover, the term *graffitiscape* identifies an essential aspect of graffiti, the nature of standing between tangible and intangible dimensions (The authors use the concept of 'tangible' and 'intangible' in relation to cultural heritage as defined by the UNESCO, https://uis.unesco.org/en/glossary, accessed on 23 July 2023). They are tangible graphic manifestations composed of signs, texts, drawings, and forms that can be captured and documented. However, at the same time, they record actions and practices that fall under intangible cultural heritage (The reference is to the UNESCO definition of 'Intangible Cultural Heritage' https://uis.unesco.org/en/glossary, accessed on 23 July 2023) elicited by the environment/context that can be recovered through their analysis. For instance, the willingness to mark a personal passage from a place or to leave a sign of worship in a shrine—as will be discussed in the second part of this contribution—has a material component, that is, the visual evidence scratched or traced on the walls. An immaterial one premises the feelings, rituals [6] (a ritual is intended as a stereotyped sequence of activities involving gestures, words and objects, performed in a sequestered place, and designed to influence preternatural entities or forces on behalf of the actors [6] (p. 1100)), traditions, and practices underlying graffiti making. They are as relevant as the material ones, and they can be recovered through the analysis of the material evidence.

For this reason, graffiti is a peculiar and original source, able to shed light on personal everyday life, practices, and traditions usually not attested by other written sources or archaeological evidence. In other words, we know that the sites with graffiti were populated and visited; however, graffiti places people back in specific spots, showing their position, the point of interaction, and the way and reasons why they physically interacted with those spaces. Moreover, the study and interpretation of graffiti retrieve the perception and interaction of past communities with the site, or, in other words, the sense of place for the past communities. (The 'sense of place' is intended to refer to emotional bonds and attachments, both positive and negative, that people develop or experience in particular locations and environments. Additionally, they used them to describe the distinctiveness or unique character of particular localities and regions [7] (p. 96)).

Graffiti, therefore, represents a crucial point in building and fostering the present sense of place by offering a diachronic perspective and bridging the past communities of users with the present ones. They create an immediate connection between past and present where visitors can physically see and 'read' the marks and messages left centuries ago by other visitors or users. They create a sense of shared experience able to overcome the time dimension, fostering the definition of a sense of place.

### 1.1. Graffiti in the Underground Built Heritage: Some Preliminary Considerations Based on the U4V Experience

In the frame of the COST action Underground4value (U4V), a crucial role has been played by the UBH case studies (for a list and description of the U4V case studies, see the dedicated section at https://underground4value.eu/case-studies/, accessed on 23 July 2023), where through the Living Lab approach, it was possible to test and develop specific tools to allow the local communities and stakeholders to valorize and use the UBH as a catalyzer for a sustainable community development. One of the aspects that were recorded but have not yet been properly addressed is the presence of graffiti. This paper aims to be a first step in bridging this gap, offering the possibility of having an additional and valuable element—graffiti—to reinforce the work of regeneration, valorization, and conservation of UBH.

The U4V case studies were selected based on their characteristics to ensure a wide variety of contexts. The result is a very representative selection of sites with a diachronic chronology, very different functions and uses, geographical distribution, and economic, social, political, and cultural diversity. Within the 16 U4V case studies, graffiti was detected in six of them (Goreme—TR, Naples—IT, Paros—GR, Ayia Napa—CY, Dolmen de Antelas—

PT, La Union—ES). In these sites, they testify to a wide variety of content and functions distributed across centuries: from the rock art of the V/IV millennium BC at the Dolmen of Antelas in Portugal, to the medieval and contemporary graffiti in Goreme cave churches, to medieval pilgrims' graffiti in Ayia Napa, to grand tour visitors in the Paros quarries, to the second World War graffiti in the underground of Naples, to the contemporary graffiti in the abandoned buildings of La Union mines in Spain.

For the remaining cases, the absence of graffiti can be an indicator of other factors related to various aspects. The absence of graffiti can be attributed to different reasons connected to the physical characteristics of the site. The state of preservation or the change in use is often the cause of a physiological loss or decay of the surfaces hosting graffiti. In other cases, the surface is not suitable to be scratched or inscribed with pigments due to its physical characteristics, and graffiti cannot be traced. This can be the case with the rough surface of a tuff or limestone cave, for instance, the Camerano caves in Italy. Moreover, environmental characteristics, such as peculiar UBH microclimates, can threaten the preservation of the surfaces, making them unsuitable for carving or tracing graffiti, as is the case of the cave church of Ayia Napa (CY), where the rock-cut walls of the cave church suffer from high humidity degradation. The original plaster is preserved in a few sections, and it is very damaged. Nevertheless, traces of a high concentration of graffiti are visible, indicating that most of the material has been lost.

On the other side, immaterial aspects, such as the activities and the UBH sites, such as, for example, the Nevère (CH) and the underground flour mills (MT), can create an environment where this kind of graphic interaction with space (graffiti-making) is not sensed, therefore does not develop. A more in-depth analysis of the U4V case study focusing on the presence/absence of graffiti and its possible relation to physical or contextual elements can provide additional information for a more accurate site description. This will contribute to a better understanding of the phenomenon in its diachronic development through the identification of possible patterns or trends across space and time in different socioeconomic and historical contexts.

Therefore, the present paper aims to be a first step in this direction, with a twofold aim. On one side, the authors want to raise awareness about the presence and relevance of this source in the context of the underground built heritage by presenting how graffiti can be documented and studied. Moreover, their contribution to adding layers of information for better knowledge and understanding of the considered sites will be extensively presented. On the other hand, they want to promote graffiti as an agent to develop the sustainable valorization, regeneration, and cocreation of the UBH sites. More specifically, referring to the promotion of UBH through sustainable practices as presented by Lo Presti and Carli [8], graffiti is relevant and can contribute to different steps of the process [8] (pp. 4–5). In the first place, graffiti can support the identification of present values, strengthening the heritage interpretation [8] (pp. 2–3) by offering a diachronic perspective on space perception and approach. Additionally, graffiti can be a relevant aspect to be highlighted within the site valorization phase since it is a visual expression left by past users. Their presence can facilitate the creation of engaging storytelling based on human presence, fostering the connection between past visitors and present ones. These aspects will be better presented in the following part and in the case studies considered.

### *1.2. Historic Graffiti: Two Levels of Addressing Sustainability*

Cultural heritage is mostly linked to the social pillar of sustainability [8,9], even though aspects of the other two pillars—environment and economy—are involved, mostly in the frame of sustainable tourism [10,11], as explained here below.

### 1.2.1. Historic Graffiti as a Sustainable Form of Visual Expression

Thanks to its characteristics, historic graffiti per se can be considered a sustainable form of visual expression related to social and environmental aspects.

*Space interaction and use of already available surfaces*

From a material point of view, historic graffiti can be considered a sustainable form of visual expression for two main reasons: from an environmental point of view, graffiti marks urban and anthropic spaces by using already existing surfaces and materials as their canvas. Therefore, instead of fabricating or building new physical supports, graffiti makers utilize every available area of existing surfaces, even creating palimpsests, such as in the case of Saint Helena Chapel, where unorganized graffiti marks can be found between organized clusters of chiseled crosses.

Additionally, all human-made surfaces used by graffiti makers were initially created for other reasons and functions. From a social point of view, their physical presence enabled graffiti makers to find a place to express themselves and record their interaction with the space. In this way, by making graffiti, the makers shape the existing space, adding new layers and new possibilities to the original functions of natural or anthropic elements. For instance, the pillars inside the Saint Helena Chapel have the structural function of supporting the dome. Graffiti makers saw in them a supplementary, blank surface in proximity to the holiest area of the church suitable to place their marks to commemorate their passage while benefiting from the proximity to a sacred place.

*Graffiti as an inclusive form of visual expression*

Graffiti is a very accessible and inclusive form of visual expression. Their creation does not require particular materials such as paper, parchment, or specific writing tools. Natural or anthropogenic spaces offer plenty of surfaces suitable to host graffiti. Simple and reachable tools, such as nails, knives, pins, or charcoal, can be used to leave a mark. Therefore, from a social point of view, everyone could potentially create graffiti without discrimination based on gender, social position, level of literacy, or economic background. Graffiti expression has no specific writing rules or patterns to be followed, representing an inclusive form of visual expression. People with no ability in writing as well as social categories usually absent from other written records could leave their marks by drawing. Sailors, for instance, have been identified for tracing very detailed ships [12]. Therefore, they are also relevant markers of social diversity and multiculturalism, as testified, for example, by the coexistence of Latin, Greek, and Armenian graffiti recorded in the Ayia Napa cave church and Saint Helena Chapel.

If we consider the phenomenon of graffiti and its long-lasting duration, it appears to be the more sustainable and durable form of visual communication. However, a clarification is needed in regard to contemporary graffiti, or better 'Street Art'. The introduction of new creative tools—e.g., spray varnishes—and techniques—e.g., stencils—modified the sustainability of the practice. The new materials are no longer freely available, and the interaction with the environment is less sustainable since the sprays damage most of the surfaces they cover. Sustainability, then, is hindered by its social and environmental aspects. In addition, the visual solutions offered by sprays and paints are more invasive, and sometimes they are used—not always and not exclusively—to express dissent, reaching at times, forms of vandalism or destruction (for the changing attitude towards historic graffiti, see Champion [13], while for the debate on contemporary graffiti, see Avraamides and Tsilimpounidi [14], Ross [15], and Vanderveen and Van Eijk [16]). A more focused and in-depth discussion about these aspects is definitely needed within the broad field of graffiti study to expand and develop the above-mentioned point, helping to better understand this practice across time and space.

### 1.2.2. Historic Graffiti as an Agent for Fostering Cultural Heritage Sustainability

In the process of promoting and building cultural heritage sustainability, the local communities play a crucial and active role in shaping the cultural heritage and in transferring it to future generations, mostly in terms of defining values to valorize and promote sustainable forms of exploitation such as cultural and creative tourism [10].

Within this framework, graffiti can be an efficient agent to foster various aspects of cultural heritage sustainability. As described above, graffiti is the repository that collects the actions, practices, memories, and values of past communities, and this can be a relevant

agent to engage and activate the local communities in the process of sustainable development. Two aspects of graffiti have been identified as particularly suitable to support this process. The first focuses on the potential of graffiti in recording and transmitting collective memory (the relevance of collective memory as an agent for sustainable development is discussed in Keramati Ardakani and Ahmadi Oloonabadi [17]). The second consists of the capacity of graffiti to testify to the interaction of past people with the monument and its landscape and, through analysis, to trace the perceptions and attitudes of past people towards them.

*Graffiti as a repository of collective memory*

As mentioned above, graffiti is a repository of collective memory on two levels: the first one is related to visual culture. Graffiti allows a very accessible and not mediated form of expression that goes beyond canons and rules and leaves the writers to choose the most appropriate form of expression—textual or pictorial. In doing so, people use and adapt elements from their visual culture and record them through graffiti. This offers us an original bottom-up insight into the everyday life and visual culture of past centuries, integrating our knowledge with information not recorded by other visual materials. The second level relates to graffiti as a physical interaction with the space. As will be extensively explained in the following paragraphs, the study of graffiti allows us to trace past people's perceptions and interactions with the surrounding space. Therefore, they will enable us to add a layer of information to the building/space by recording the memories of past activities and practices that, most of the time, are not otherwise attested.

*Graffiti as a catalyzer of community engagement and sense of place*

Graffiti acts as a mirror to the past, a visual layer that connects present visitors with past ones. Moreover, graffiti offers a different approach to the building, not following the top-down structural planning but pointing out alternative ways of experiencing that space. Visitors to religious shrines, for instance, are typically guided to interact with the surrounding space through its architectural characteristics, which developed to be functional for the rituals and practices of a supervised and formalized system. Attention is usually conveyed to the more relevant parts connected to the liturgy or specific cults (e.g., the altar or relics). Graffiti does not always reflect this spatial convention. By recording people's experiences and perceptions of the space, they create an alternative path, which may or may not correspond to the official one.

Therefore, when graffiti is present on a site, its study and dissemination can be a relevant element in engaging the present community by sharing how past generations experienced those places. Moreover, since graffiti expresses the perception and interaction of past people with that space, these contents can be used to foster and promote a sense of place from a diachronic perspective. Present communities can learn about past values recorded on the place through graffiti and start a reflection concerning the present values the site has now. This represents one of the key steps for developing sustainable approaches to the exploitation of cultural heritage through cultural and creative tourism [10,11].

## 2. Underground Graffiti in Context: Two Case Studies

This part is devoted to showing a practical application of graffiti documentation and study through two examples: the cave church of Ayia Napa monastery in Cyprus and Saint Helena's chapel in the Holy Sepulchre in Jerusalem. (The specific methodology and the digital tools and approach used for the documentation, cataloging, and study of graffiti are presented at the end of the paper in Appendix A. Therefore, this part will present the graffiti interpretation).

The two case studies have been selected in a critical way, considering the characteristics of the sites, first of all the fact that they are both underground parts of more extended buildings that later developed above ground. Other crucial aspects include the fact that both are Christian religious buildings with a very active and vivid cult. Moreover, both sites attracted different Christian communities within the same space [18] (pp. 443–444).

On the other side, differences between the two sites are present and functional to understand how the same phenomenon—the making of graffiti—developed in both sites, fostered by devotion and worship but in peculiar and original ways based on the specific characteristics of each context.

All these aspects will be described and highlighted in the following part to allow a better understanding of how devotion and worshiping practices were expressed in the past through the making of graffiti, producing original variations based on the material and immaterial characteristics of the site.

Moreover, something that needs to be clarified is that the two sites have been studied in regard to graffiti presence independently; they are not part of a common project, and the research questions were different. In neither case were the sites approached in light of the sustainability aspects presented here. In this regard, the two cases chosen are particularly relevant, as they present two different stages of the research. In the case of Ayia Napa, the documentation and study of graffiti are already included in a museum planning project that includes graffiti as an indicator of the past sense of place. The museum concept foresees the use of graffiti as an engaging element to involve the local community as well as visitors and tourists. In Saint Helena Chapel, the focus was on understanding and dating the presence of crosses in relation to other graffiti and to the site, presenting a very first step towards the acknowledgment of their presence and their historical and social value. The exploitation of graffiti and crosses as agents for the sustainable development of this space is still in progress; some possibilities will be suggested in light of what emerged from the research, but nothing concrete is planned for the moment. The structure of this part, therefore, summarizes the two research studies carried out and shows how, in both cases, graffiti can become an agent to foster UBH sustainability.

## 2.1. The Cave Church of Ayia Napa, Cyprus: Research Background

The study of the graffiti of the church and monastery is part of the research activities concerning the material and immaterial aspects of the site in the framework of a broader project for the creation of a museum in the complex (Figure 1). The action is promoted by the municipality of Ayia Napa and the Bishopric of Constantia and Famagusta, an international team led by Prof. Brigitta Schrade of the Freie Universität of Berlin and supported by members of Ca' Foscari University of Venice and the Cyprus Institute STAC—APACLabs. The project will deliver the structure and content to be implemented in the rooms around the courtyard. The aim is to highlight the role of the monastery in the local community and foster its relevance as a catalyst for the sociocultural and economic development of the village of Ayia Napa and all of the area. Graffiti will play a relevant role within the process of identifying values and engaging the local community in the definition of the sense of place, starting from the experiences recorded on the walls by past worshippers and visitors. This process has been supported and helped by the COST action U4V, which crucially contributed to shaping a strategy to enhance the sustainable impact of the museum. By highlighting the relevance of the UBH as a sociocultural and natural element and promoting its knowledge, preservation, and promotion, the U4V COST action helped to shape a more sustainable touristic offer. The work conducted is presented in the Second Handbook of U4V [19], while the volume on the case study and Living Lab developed in Ayia Napa is under preparation.

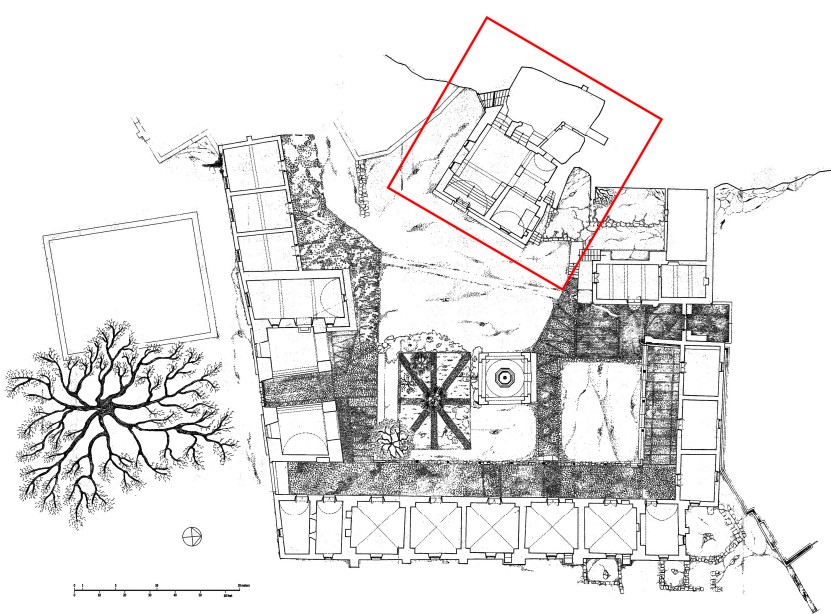

**Figure 1.** Plan of the Ayia Napa monastery complex with the area of the cave church highlighted ([©] Andrea Economou).

### 2.1.1. The Site and Its Origin

The cave church of Ayia Napa is the main historical landmark of the homonymous village and its area. Located on the southeastern coast, the site is part of the natural landscape and is characterized by a limestone formation rich in natural caves and freshwater springs. The geological conformation of the area has offered, since prehistory, natural caves to be inhabited [20]. Since then, the underground natural spaces have always been integrated with the everyday life of the inhabitant, fulfilling different needs, from the supply of building materials (the quarry of Makronissos) to funerary (Makronissos funerary complex) and religious functions (cave churches are very popular in the area, see, for example, Ayia Napa church, Agia Thekla, Agia Marina, and Agii Saranta), to shelters for sharpers and their animals (these spaces are still visible in the area of Paralimni/Protaras).

The cave from which it originated and developed the church and monastery of Ayia Napa perfectly reflects the diachronic use of the natural cave to fulfill local needs, fostered by the presence of a natural spring collecting the water from the hills around (Figure 2).

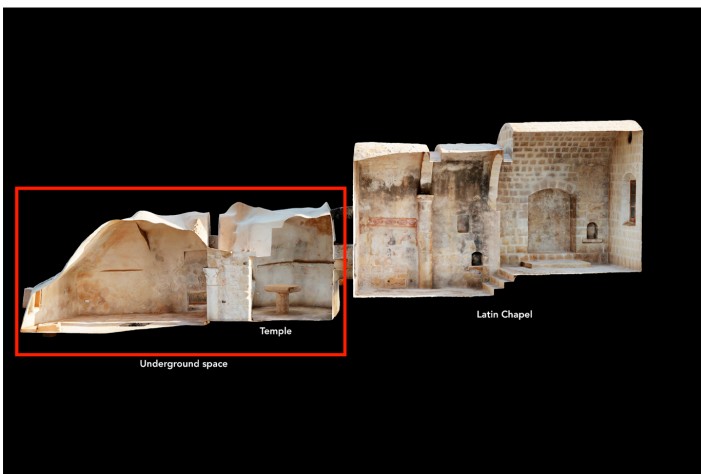

**Figure 2.** Church of the Ayia Napa monastery. (Photogrammetric model by Theo Lerle).

The site originated with the miraculous finding of an icon in the early Byzantine period. According to tradition, the icon was found by a man digging to build his house.

After the miraculous discovery, it started to be venerated in the cave [18]. Since the very beginning, the icon showed miraculous powers; the site grew in relevance, and the cave was adapted to the orthodox cult by defining the area of the *bema* (temple) with a built-in iconostasis. A further addition was made during the Lusignan period (1191–1489), adding a chapel on the east side of the temple. During the Venetian domination (1498–1571), the building consolidated its present form with the addition of the main entrance and hall, guiding the visitor through a staircase to the underground part of the complex (Figure 3).

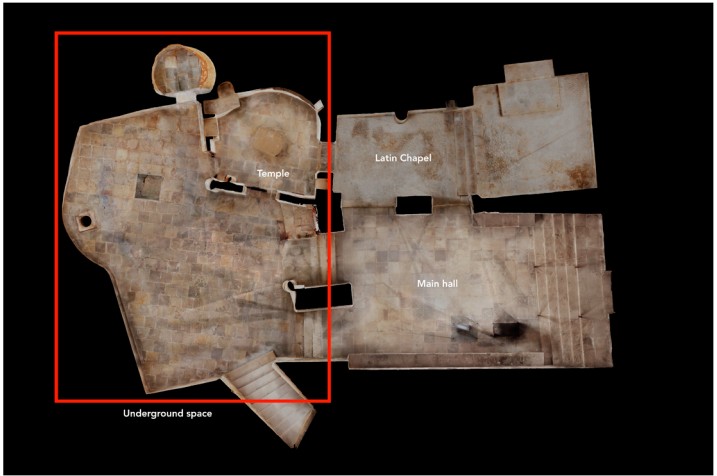

**Figure 3.** Plan of the church. (Photogrammetric model by Theo Lerle).

Numerous pilgrimage accounts tell us about the great importance held by the site during the Lusignan and Venetian periods, hosting the Orthodox and Catholic communities in a shared shrine [21]. While the Orthodox were celebrating in the lower part, the proper cave, the Latins were allowed to carry on their religious functions in the 'Latin chapel', which had access to the temple thanks to a big window cut into the masonry.

Numerous pilgrimage and visitor accounts, mostly recorded between the 15th and 17th centuries, describe the icon of the Virgin, highlighting the cult expressed through plenty of τάματα (*tamata*) and ex-votos. The miraculous power of the icon was reinforced by the presence of the αγιασμα (agiasma—Holy well), with curative water.

It is in this crowded and lively context that graffiti started spreading in the building, leaving a tangible on-site mark of the passage and devotions of worshippers.

2.1.2. Graffiti in the Church Complex: Spatial Distributions and Types

The graffiti distribution has been represented through influence and heat maps layered on top of a floor plan of the building (Figure 4a,b).

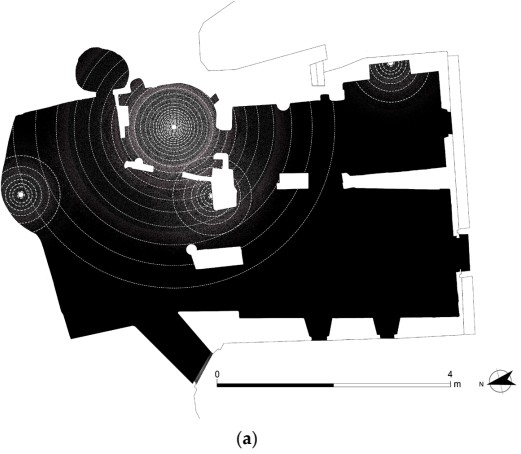

(**a**)

**Figure 4.** *Cont.*

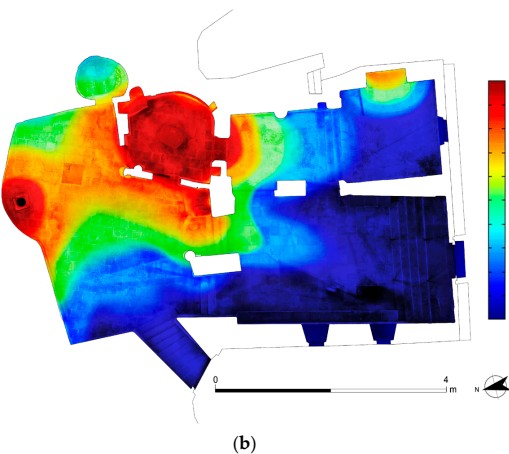

(**b**)

**Figure 4.** Ayia Napa monastery church: (**a**) Sacred influenced; (**b**) Heat map of graffiti distribution and concentration. Concept and graphics Andrea Tinazzo$^©$. The colors represent the graffiti distribution: red represents high concentration and blue represents low concentration.

The highest concentration is attested in the north chapel. The lowest concentrations are present in the entrance hall and in the cave church—west wall. However, this picture is misleading. While it records the exact distribution of graffiti, it reproduces what is visible and preserved today. In fact, while the walls and surfaces of the north chapel and hall are generally well preserved, with the plaster still intact where present, the situation is drastically different from what concerns the cave space. Here, the present plaster is the result of recent restorations that left in place a few fragments of the original one, on the west and northwest sides of the cave. The fragments are less than 2 m high, and they preserve traces of graffiti in high concentration. From what it is still possible to see, they are all Greek inscriptions traced with a black pigment that, unfortunately, due to the small section preserved and the bad state of preservation, are difficult to read and reconstruct. Most likely, the entire surface of the cave space, at a height suitable to be inscribed (usually between 1 and 2 m high) was originally covered with graffiti, as indicated by the two surviving fragments. Unfortunately, there is no way to reconstruct the original appearance, as there are no pictures known so far showing the site before restoration or written records mentioning the presence of graffiti. Despite this, it is necessary to bear in mind that the presence and distribution of graffiti were initially different from the current ones, at least for what concerns the cave church. The north chapel appears to have best preserved its original appearance. The space is divided into two areas, separated by a few steps. The part next to the altar (west side) has the walls covered with plaster, while the other one has the stone of the masonry exposed and was probably never covered by a coating. All the graffiti present in the north chapel and in the hall is displayed at a height between 1 and 2 m approximately, with the exception of some inscriptions on the west wall of the chapel, which are located between 2 and 3 m in height. To create this graffiti, the writers most likely used pedestals or a ladder, as there is no information or evidence of a variation in the floor level.

The still-visible graffiti testifies to the presence of both textual and pictorial forms. The texts are in three alphabets, Latin, Greek, and Armenian. The Latin alphabet records texts in Latin, Vernacular Italian (Venetian), and French. They record names, initials, dates, monograms, and, in one case, a more extended text still under study.

The pictorial graffiti includes a ship, a coat of arms, and identity marks made up of letters and geometrical elements. The graffiti is mostly traced with black pigment, while very few have been scratched directly on the masonry.

### 2.1.3. Graffiti Documentation

The documentation of graffiti represents the first step of their study, and it needs to be performed on two levels: starting from the monument scale and shifting to the graffiti one.

The whole church has been documented through photogrammetry by Theo Lerle in the framework of the research project for the musealization of the site. The photogrammetric model has been used to locate the graffiti, highlighting their distribution in a three-dimensional, lifelike space. This operation has a twofold aim. On one side, it locates the graffiti in the space, identifying their position in relation to the different parts of the building. The spatial distribution, as explained, is crucial for the final interpretation of graffiti based on its position (space) combined with its content and form (Appendix A). On the other side, it supports the holistic documentation of the building, highlighting the presence of graffiti visualized in context.

Afterwards, graffiti was documented, focusing on the north chapel, the best-preserved area with the highest concentration of material. In this area, graffiti was realized with two techniques: by scratching with a pointed tool the plaster or directly the masonry, or by tracing with black pigment letters and drawings. Carved graffiti presents quite deep furrows, making them well visible and readable with traditional photography, sometimes supported by the use of raking light (RAK). This technique uses a source of light placed at an oblique angle to the surface documented, highlighting the irregularities, incisions, and surface texture. Carved graffiti becomes more visible, and small details, difficult to see in normal light conditions, can be detected (Figure 5).

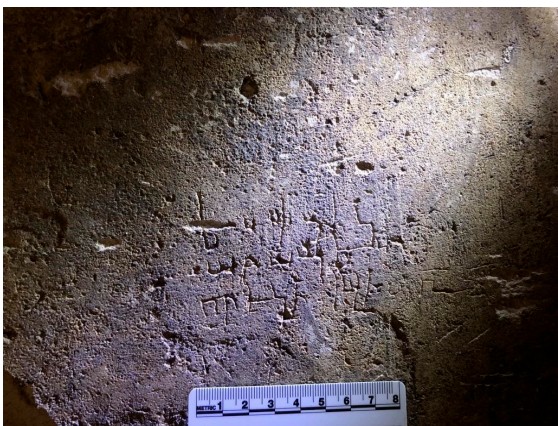

**Figure 5.** Armenian graffito highlighted through Raking Light technique (Photo: M. Trentin).

For traced graffiti, the documentation has been more challenging. The aged inscriptions were hardly visible to the naked eye due to pigment loss. We used infrared imaging (IR) to better visualize faded parts of the inscriptions that are otherwise not visible to the naked eye. Thus, this technique allowed us to refine the preserved fragments of the wall graffiti as well as visualize the faded ones, improving their readability (Figure 6a,b).

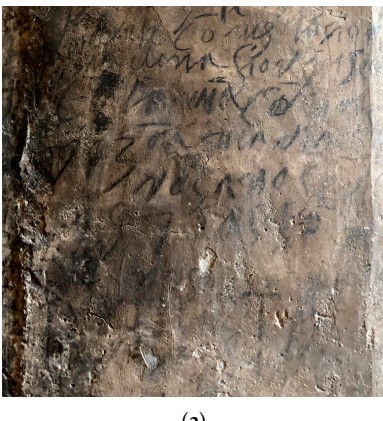
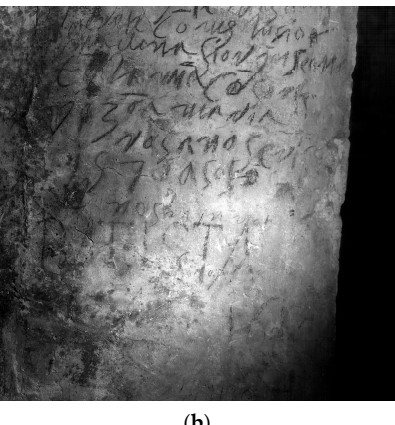

(**a**)                    (**b**)

**Figure 6.** Latin chapel—North wall, traced graffiti enhancement: (**a**) Traditional photography; (**b**) Infrared Imaging (IR) by Svetlana Gasanova.

2.1.4. Graffiti Interpretation

The graffiti collected in the documentation campaigns is still under study, but preliminary results already highlight relevant information. Most of the graffiti preserves names and dates, initials, and identity marks, with Latin inscriptions prevailing over Greek and Armenian ones. Their form—Dates and name or identity mark—Indicates the commemorative function and, therefore, the will to record the visit to the shrine. Their location, though, also indicates the will to interact with the sacred (Figure 7).

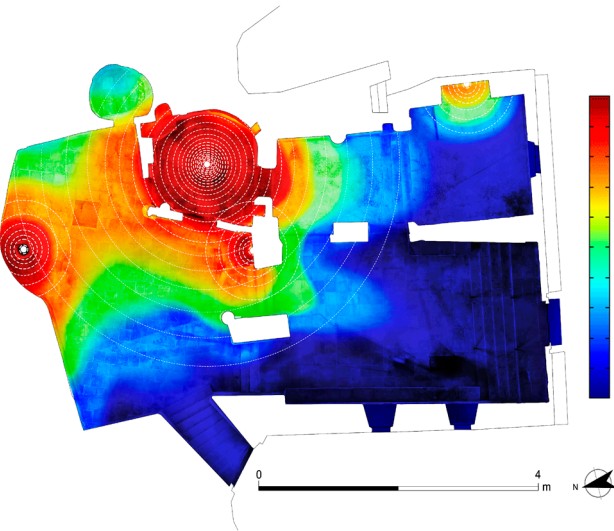

**Figure 7.** Graffiti distribution in relation to sacred space (sacred influence map and heat map of graffiti distribution combined). Concept and graphics Andrea Tinazzo©.

Only a couple of inscriptions, in fact, have been detected in the hall, traced with black pigment. It seems that the writers concentrated their activity in the north chapel, the one directly connected to the temple—the most sacred place in the building. While some inscriptions are high above the window, the majority are traced on the north wall, on the side of a fresco depicting three female saints [18] (p. 443). The Latin and Greek texts can be attributed to a chronological gap between the end of the 16th and the beginning of the 18th century, as also attested by some dates. These graffitis are commemorative in nature but express, at the same time, devotion and the will to connect with the sacred. Leaving a mark near the fresco with the saints supports with tangible evidence what the pilgrims' and travelers' accounts narrate about the popularity of the site.

A particularly interesting graffito is located on the east side of the chapel, on the left side of an altar obtained in a recess of the wall on the north side. The graffito, carved with precision on the tuff stone, records '*Deusedit/hic celebravit/October* (*October* is written with the numeral *8* followed by the *b* with an abbreviation sign on the stroke) *28*' (Deusdedit officiated here on the 28 October 1777). Despite the architectural form of the altar, its position in a marginal area of the building has always left open the question of its actual function. This inscription confirms that, for sure, in the second half of the 18th century, people were officiating liturgies, quite surely according to the Latin rite.

Graffiti in the cave church offers new insights into the everyday life of the church, highlighting the vivid cult of different communities, the Orthodox and Catholic ones, also joined by Armenians, as attested by a few inscriptions. In the cave church of Ayia Napa, the architectural structure developed to accommodate the multicultural Lusignan and Venetian society of the island. Its walls, covered in graffiti, became silent witnesses to shared practices of worship and devotion. Unfortunately, the graffiti that probably covered the cave church plaster has been lost, preventing comparison with the ones preserved in the north chapel. Nevertheless, the few fragments preserved testify to their presence, and hopefully, future documentation campaigns will be able to record what is still preserved.

2.1.5. Graffiti as an Agent for Promoting Sustainability in the UBH of Ayia Napa

The results of the documentation and study of the graffiti, specifically in the cave church, are included and integrated within the broader structure of the museum, currently under definition. The general concept of the museum is to re-connect the site with its landscape, crucially modified by a massive and uncontrolled development in the last decades, functional to a 'resort' model of tourism that is now outdated and not sustainable. The requalification of the complex and its revalorization as a museum—in a structural and content way—Aims to create a first and relevant landmark to start the discussion and possible adaptation of the current touristic offer.

With the support of the U4V COST action, the local stakeholders and community have been engaged, and the present planning phase shows good potential for what concerns the recovery and valorization of the UBH sites, of which the area is very rich.

The contribution of graffiti within this broader frame has been identified as a relevant agent to foster the sustainable development of the site. More specifically, graffiti studies recovered the past collective memory and sense of place, fostering the present religious values attributed to the place by the present community with a diachronic and multicultural perspective. Moreover, the practice of leaving a mark, a personal trace of the passage, is something that is still happening through the practice of 'tamata' (τάματα), small gifts like silver lamina reproducing a part of the body or shaped candles, or even small religious bracelets Κομποσκοίνι (https://en.wiktionary.org/wiki/komboskini#:~:text= komboskini%20(plural%20komboskini),recitations%20of%20the%20Jesus%20Prayer, accessed on 23 July 2023), left to ask for specific protection or just to set the presence in that holy place. Tracing the roots of this present practice back to graffiti-making highlights the long-lasting tradition of worship and devotion experienced by the site, at least since the 14th century. Currently, these aspects are elaborated through storytelling to raise awareness among visitors about the graffiti's presence and significance and to share with them the value and history of the place as a sociocultural and historical landmark in the area. Thanks to digital tools (the plan is to use the photogrammetric models developed by Theo Lerle to create virtual tours enriched with dedicated contents such as enhanced imaging for the graffiti, short videos with the bishop introducing the site, and different specialists talking about the cave church highlights (e.g., the icon, the architectural structure, the graffiti), graffiti and other elements inside the church will be largely accessible thanks to virtual tours, ensuring accessibility to the information and, at the same time, the preservation of the site in its religious and historical aspects. No information board or visual material will be displayed inside the cave church to preserve its religious character and function.

Despite the on-going work, graffiti has contributed, so far, in the operational phase [8] by providing elements for the heritage interpretation supported by the local community and stakeholders. Thanks to this discussion, the local stakeholders, mostly engaged in the tourist economy of the area, started considering the benefits—so far mostly from an economic point of view—of recovering the tangible and intangible cultural heritage of the area as a mitigation measure for the high degree of seasonality. In addition, the concept of creative tourism, with dedicated experiences linked to the rural life of the area, has started to be developed with success. In this frame, the UBH that characterizes the natural and sociocultural landscape of the area can be a catalyst to promote sustainability in cultural heritage through sustainable forms of exploitation, such as creative and cultural tourism.

In this perspective, graffiti, as argued, can support the process and offer engaging elements to easily connect people with the space and its history.

*2.2. Saint Helena Chapel—Jerusalem: Research Background*

The chapel of St. Helena in the Holy Sepulcher of Jerusalem is an underground structure leading to a subterranean cave built by the Crusaders on the former remains of the great Constantine Basilica from the 4th century [22] (pp. 11–14). Tradition holds it to be the place where the empress Helena, mother of Constantine, found the true cross (Figure 8).

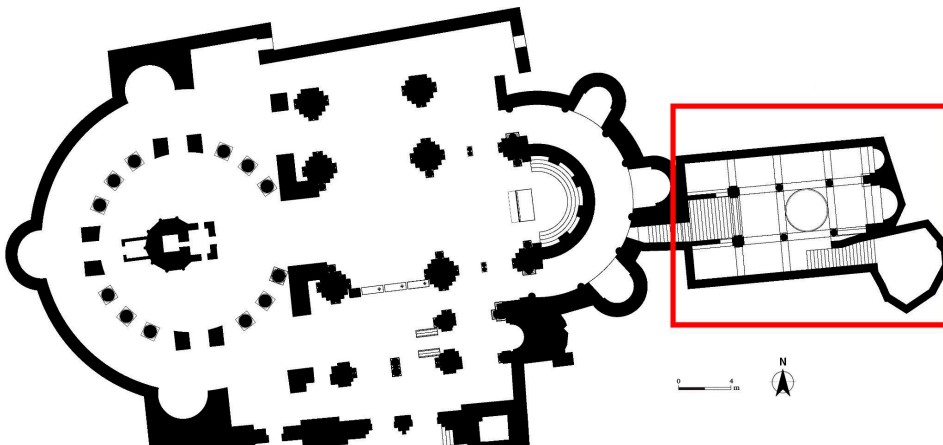

**Figure 8.** Complex of the Holy Sepulchre with Saint Helena Chapel highlighted (Or Roz and Amit Re'em drawing).

Walking down the stairway leading to the underground chapel, one immediately encounters the most striking phenomenon of the chapel, numerous rows of crosses neatly engraved into the walls on either side, which thereupon extend by their thousands over the entire wall surface, especially within the apses behind the altars.

Popular opinion has attributed these crosses to the graffiti of pilgrims over the centuries, dating back to the Crusader period [23] (pp. 101–102) [24] (pp. 93–94) [25] (pp. 5, 104). However, the research led by Dr. Amit Re'em of the Israeli Antiquities Authorities (IAA) [22,26] has called into question this assumption and consequently set several goals:

- To date the Crosses, that is, to refute or confirm the perception that they are from the time of the Crusaders;
- To consider if the crosses can be categorized as graffiti;
- To decipher the identity of the cross-makers;
- To understand the phenomenon and its functions.

In order to accomplish these goals, the first step was to provide high-quality and focused documentation of the crosses and their context. To achieve that, a variety of digital imaging techniques and approaches were deployed. Photogrammetry, laser and structured light scanning, reflectance transformation imaging (RTI) (Figure 9b), gigapixel imaging (Figure 9a), LIDAR, etc. The different techniques were selected and applied, taking into consideration the research needs and the site/context characteristics, such as accessibility and light conditions. For example, due to the sensitivity of the place, it was a given that one had to employ a variety of quick, noninvasive techniques to gather as much information as possible for later analysis.

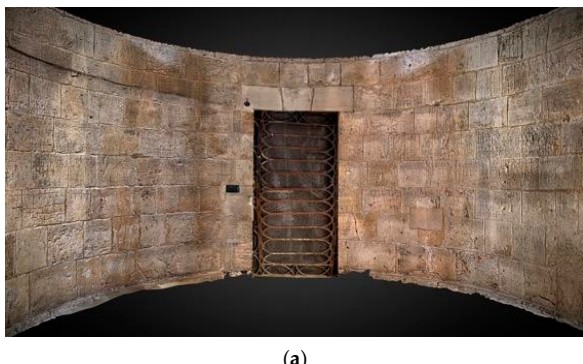

(**a**)

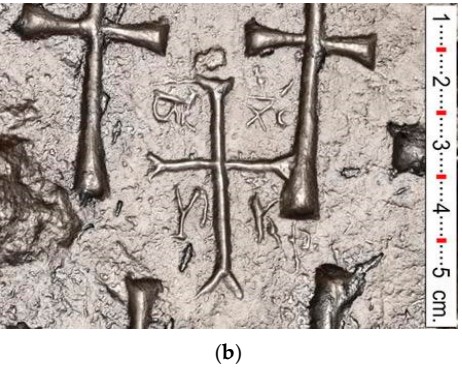

(**b**)

**Figure 9.** (**a**) Gigapixel of the apse with crosses (Photo: M. Caine and D. Altaratz); (**b**) RTI detail (Photo: M. Caine and D. Altaratz).

Different techniques have their own advantages and disadvantages when it comes to interactive control, lighting, measuring, filtration, magnification, and other factors [27,28]. For instance, when faced with the challenge of documenting and evaluating hundreds of stones on a wall while still being able to zoom in for detail, Gigapixel imaging was used. On the other hand, RTI was necessary in situations where high-quality surface information was needed for texture analysis and the identification of faint surface details.

### 2.2.1. Dating the Crosses

The first challenge encountered after the documentation was the identification of the crosses' chronology. The key point was the comparison between the crosses and other datable graffiti located between them. A first useful example was a graffito located in the lower corner of the southern apse, recording a heraldic symbol with an inscription bearing a name (Figure 10a,b). The symbol was a simple shield divided in half by a left diagonal line. The inscription was written in capital letters, showing an Italian influence ascribable to the third quarter of the 15th century in Germany, indicating the name M. V. WILDEN/STEIN RITER (M(artin?) V(on) Wildenstein knight) [29] (p. 126) [24] (p. 183) [22] (pp. 21–24). This was traced to a family of imperial knights from Middle Franconia. A member of this family, Martin Wildenstein, made a pilgrimage to Jerusalem in 1503 and was dubbed a knight of the Holy Sepulcher on the third of October. He probably engraved his name alongside his coat of arms. Close inspection showed that the crosses were carefully carved at the lower part of the stone and on the nearby stones in a way that would not harm the coat of arms or the inscription. Hence, it would appear that they were engraved while the symbol and the name already existed. Therefore, the crosses could not predate the 16th century.

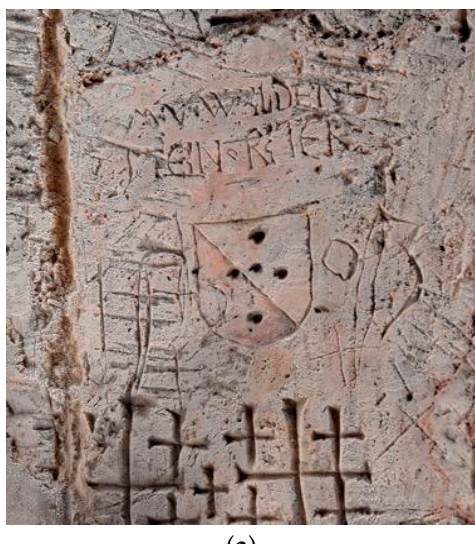
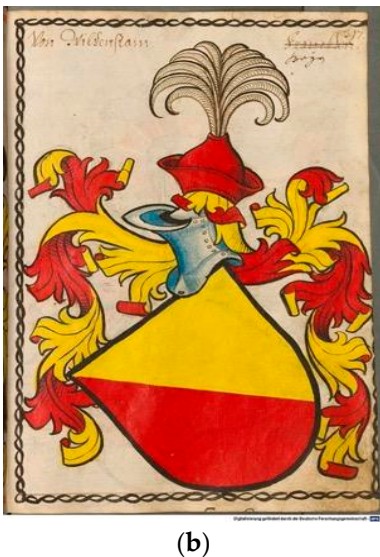

(**a**)                    (**b**)

**Figure 10.** (**a**) Graffito with Wildenstein's coat of arms (Photo: M. Caine and D. Altaratz); (**b**) crest of the Wildenstein family from the Scheibler Armoria.

Moreover, on the lower part of the main corridor leading to the chapel, on the southern wall, another heraldic sign was located. Already identified by Detlev Kraack [29] (p. 125) and archaeologist Jürgen Krüger [24] (p. 183) [22] (pp. 24–26), it consists of the upper part of three hearts above a wall that is built over three arcades (Figure 11a,b). Despite not revealing the owner's name, something no less important was apparent. The RTI of the symbol showed that it was surrounded by a thin rectangular frame that had worn down over the years (Figure 11a). At the bottom of the frame are Roman numerals that make up the date. It is important to note that the frame, the date, and the symbol are one piece/one creation. The date itself consists of three letters. Maybe one or even two additional letters were missing as they were cut by the arm of a cross. Assuming the missing letter is D, the date is 1520, as suggested by Detlev Kraack [29] (p. 125). Possibly also another letter, L was

to the right of the D, and may suggest the date 1570. Either way, the very fact that the date and the frame were cut by the crosses indicates that the crosses succeeded the Roman date, and therefore, at least concerning this concentration of crosses, they are not earlier than 1520.

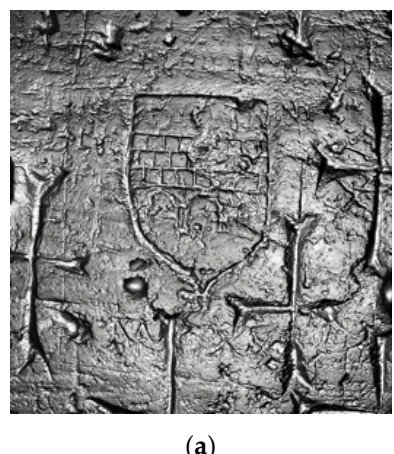

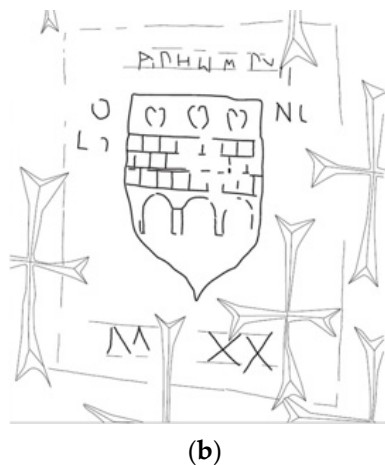

(**a**)　　　　　　　　　　　　　　　　　　　(**b**)

**Figure 11.** Unknown coat of arms: (**a**) RTI (M. Caine and D. Altaratz); (**b**) drawing (O. Zakaim).

On the southern wall of the corridor leading to the chapel, two Latin inscriptions were located. The first records the name of an Italian pilgrim from Verona—*Iustinus Veronensis*—And can be palaeographically attributed to the 15th century [22] (p. 28). The second inscription bears the name of an Italian friar from Lucca and a date—*Fra(ter) Cristoforo di Luca 1600* [29] (p. 125) [22] (p. 28). In both cases, the inscriptions are surrounded by crosses without any overlapping, testifying that the graffiti was present when the crosses were carved, offering this area a *terminus post quem* for the cross carving between the 15th and 17th centuries.

Moving to the walls of the apses, 30 Armenian inscriptions were identified. The majority of them record names that are chronologically ascribable between the mid-15th and 18th centuries based on paleographic characteristics [22] (p. 29). All the inscriptions are surrounded by crosses that respect them with no overlapping, indicating, once again, that the texts precede the carving of the crosses, ascribable in a moment after the 17th century.

The names traced in the apse, one of the main sacred areas of the chapel, can be interpreted as commemorative-devotional graffiti. Worshippers were recording their names and fixing their presence in that sacred space with a twofold aim: to record their visit and ensure the constant benefit of the sacred location of their token.

### 2.2.2. Graffiti or Not

Another research question behind the study of the crosses was the possibility of including them in the category of graffiti. Despite scholars still debating about a possible definition of graffiti [2], in this case, two characteristics observed in the groups of crosses raised doubts about their categorization as graffiti. The first concerned their accuracy and uniformity, and the second their carvers.

The physical characteristics of the crosses, in fact, indicate an accurate, planned, and organized action, which is usually not present in graffiti. The entrance portal to the Church of the Holy Sepulcher, for instance, bears an excellent example of medieval graffiti features. Hundreds of engraved crosses and inscriptions were made, including personal names, prayers, memorial inscriptions, and dates—in a variety of languages: Greek, Latin, Arabic, Armenian, Georgian, Syriac, and Slavonic. Everything is conducted with various techniques: cutting, scratching, painting, inking, or tracing charcoal onto the stone (Figure 12).

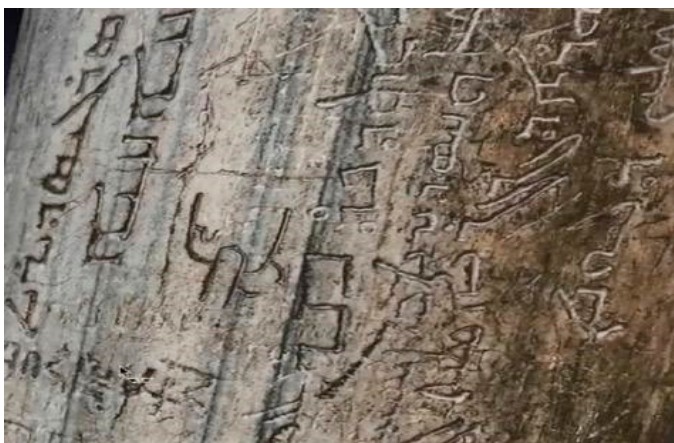

**Figure 12.** Graffiti on one of the left columns of the Holy Sepulcher entrance (photo: M. Caine).

Compared to the graffiti on the facade of the Holy Sepulcher, it is arguable that the cluster of crosses in St. Helena Chapel does not fall under the graffiti category. The crosses are neatly and carefully created in a structured distribution; there is planning behind them, and they are performed with one technique—cutting into the stone. Beyond the above-mentioned evidence, macrophotogrammetry and RTI were employed to map the crosses, based on the research of Dr. Lindsay MacDonald of UCL [26]. Examining their sizes, the depth of the engravings, and the slopes of the arms of the crosses, it was concluded that most crosses fall into one pattern. They have the same morphological characteristics. This led to the conclusion that no more than several hands created the crosses, with patterns and techniques known in advance. Hence, the engraving of the crosses cannot be seen as random graffiti.

2.2.3. Who Were the Cross-Makers?

According to Dr. Yana Tchekhanovets' research, evidence suggests that Armenians were the creators and masterminds behind the design of the crosses [22] (pp. 30–35). She highlights different elements to support her hypothesis, focusing on the preeminent role of the symbol of the cross in the Armenian tradition across centuries and territories. Despite the fact that traditional written sources do not record any reference to cross carving in religious sites, carved crosses very similar to the one visible in Saint Helena Chapel are detected in other sites owned by Armenians in the Holy Land, such as St. James Cathedral, St. Savior Church on Mount Zion, Dair az-Zaituna, Church of Nativity, and more [22] (p. 30). A unique graffito in Saint Helena Chapel offers us a relevant indication of this practice. The inscription, surrounded by crosses, records in uncial Armenian script, '*This holy cross is a memorial of Bac'in*', indicating that a cross was carved or taken as a memorial token by Bac'in [22] (p. 33). Similar graffiti inscriptions connecting the cross to a memorial token have been found in the Monastery of the Holy Archangels in Jerusalem [30] (p. 116).

Another piece of evidence Dr. Yana Tchekhanovets presents to support the idea that Armenians could be identified as the cross carvers is that the same phenomenon of cross-carving in religious buildings appears in Armenia or Armenian structures in other territories, such as Cyprus (Armenian church of Famagusta) [31] (pp. 125–141). This can be explained by the central role attributed to the symbol of the cross in the Armenian tradition through liturgies, feasts, and visual traditions [22] (p. 34).

A last point reflects on the Armenian ownership of Saint Helena Chapel—known as Saint Gregory Chapel in the Armenian tradition. According to Armenian records, the presence and ownership of parts of the Holy Sepulchre by different Christian communities happened during the reign of Saladin (1137/38–1193) [22] (p. 35). The crosses, as discussed above, are dated no earlier than the 14th century, when the Armenian community was already actively in place, at least for a couple of centuries.

Moreover, the 14th century was also the moment graffiti started spreading in the Holy Land, consolidating a phenomenon already existing but fostered by the increasing flow of pilgrims and worshippers [22] [29] (p. 36). Written sources testify that graffiti making was a popular activity to carry on as part of the visit and pilgrimage, within other practices, as attested for instance by Felix Fabri in his account dated to the last decades of the 15th century [32] (pp. 86–87) [22] (p. 37).

The central importance of the cross in the Armenian tradition and the fact that the chapel stands on the place where the tradition locates the discovery of the True Cross by Empress Helena seem to be strong elements supporting the possible identification of the cross-carvers with members of the Armenian community. Moreover, considering the high quality of the carvings and the identification of several hands behind the clusters, it is possible that stone masons were employed by the Armenian clergy to offer a paid service of professional and authorized cross-carving.

A future and more in-depth survey of the chapel, exploring the presence and chronology of other graffiti, such as the ones in the chapel's pillars, will offer an interesting insight into the coexistence of an 'official' and 'regulated' activity of cross-carving controlled by the Armenian community and more traditional self-made graffiti.

### 2.2.4. Graffiti and Crosses: Interpretation

Saint Helena Chapel represents a unique case to explore the interaction of past communities with the space and recover the sense of place of the past centuries.

The graffiti identified and studied so far records the widespread practice of leaving a mark to record a visit to that specific place. Latin and Armenian inscriptions bearing names and sometimes dates are commemorative graffiti, and their location—on the walls of the staircase leading to the chapel and on the apse—add a devotional component to that, displaying a mechanism similar to the one detected in Ayia Napa. Here too, it seems that the Armenian community had privileged access to the place since only Armenian graffiti has been found on the apse, a very prominent sacred area that is for sure less accessible compared to the walls of the staircase. Once again, graffiti in the chapel testifies to the will to leave a sign, mark the passage, and connect with the sacred.

Besides traditional graffiti, an original phenomenon was detected: cross-carving. Professional hands from the 15th century carved numerous crosses, following patterns and showing a structured and planned action. The interpretation of this activity starts with the more popular practice of graffiti making but transforms the action into a more formal and planned activity, most probably controlled by the Armenian community. The crosses are not carved as decoration; they have been created using an already available surface not intended for writing—as graffiti does. Nevertheless, the evident planning raises doubts about their possible identification as graffiti. The interesting aspect is that the presence of graffiti and carved crosses side by side testifies to two different approaches to interaction with the space: the first following a bottom-up approach and the second a top-down one. Graffiti inscriptions are nonregulated and managed by the individual who made them, interacting individually with the space. The crosses testify to the will of visitors to interact with the space, conveyed and organized by the authority of owning the chapel—The Armenian community.

Both practices are accepted and coexist; they even express the same functions: leaving a personal mark to commemorate the passage in a sacred place with devotional functions. Nevertheless, the presence of a structured and organized way of interaction, supported by the Armenian community, is a further confirmation of the importance of that specific area within the Holy Sepulcher and of the relevance of the cross as an identity symbol for the Armenian community and for the worshippers visiting that place.

### 2.2.5. Graffiti as an Agent for Promoting Sustainability in the UBH of Saint Helena Chapel

The study of the graffiti in Saint Helena Chapel has been conducted exclusively to address the research questions presented above, namely, to date the crosses and establish

their nature and relationship with other graffiti on the site. The study, in contrast with the case of Ayia Napa, was not part of a broader project of valorization or musealization of the site. The complex sociopolitical and religious context—Being part of the Holy Sepulchre site, the main Christian shrine shared by various communities—and the massive number of people visiting the chapel every day are elements that have prevented any initiative toward the sustainable development of the site. If this is the case in the future, graffiti, as argued above, can be a valuable element in supporting the sustainable development of the site by recovering aspects of the collective memory and the past sense of place.

Despite the fact that there are no plans to address aspects linked to the sustainable conservation and valorization of the site so far, the site well represents a significant number of UBH cases that are very active in their original functions. Saint Helena Chapel is a very active and deeply felt worship place within the Holy Sepulcher complex. The focus on its religious value veils other aspects, such as its undiscussed role as a cultural heritage monument (inscribed on the UNESCO list with the city of Jerusalem). This fact shifts the focus from cultural heritage matters to socioreligious ones linked to its functions. Basically, issues concerning the site's sustainable development, conservation, and valorization came after any other issue related to its function as part of one of the main Christian shrines.

In cases like this, where the process of addressing sustainability is yet undefined due to different elements. In our case, the complex nature of the communities involved and the sociopolitical situation. It is even more relevant to raise awareness and document the presence of agents that can promote and activate a sustainable approach to the site.

The graffiti of Saint Helena Chapel, now recorded and studied, is available not only to the academic community but also to the local one and stakeholders if at any point they intend to pursue it.

### 3. Conclusions

With the two case studies, the authors first aimed to raise awareness about the possible presence of graffiti in the UBH context. Graffiti, when present, must be considered and approached as an integrated element of the site, able to recover a peculiar human dimension made of everyday practices and activities otherwise neglected and not attested from other more traditional sources.

Through an interdisciplinary study, the paper illustrated how graffiti can be approached, documented, and analyzed to release its full potential as information. This can contribute not only to a better understanding of the site but also to its social dimension. The two case studies presented are relevant because they show how two religious' spaces, specifically Christian sites, were perceived and approached by their visitors and how people interacted differently based on the feelings and meaning the two sites had for them.

Moreover, the paper argues that graffiti can be a valuable agent to support the sustainable development of the UBH. Graffiti is a repository of collective memory and contributes to tracing the past sense of place. As discussed in the paper, these two elements play a relevant role in two phases of the process. Their contribution to shaping the sense of place and the value attribution to the sites where they are present can be an additional resource in the process of regeneration, promotion, and valorization of the UBH. Past and present communities share the same space, and graffiti can be an engaging bridge to connect them.

This paper represents a starting point in the process of raising awareness about the potential of graffiti as a vector of original information about UBH sites, recovering a human dimension otherwise not attested. A future and more systematic study of graffiti embracing different types of UBH sites across Europe can contribute to tracing possible patterns, providing a better understanding of human interaction, use, and functions across centuries providing, at the same time, a valuable support to foster UBH sustainable development.

**Author Contributions:** Conceptualization, M.G.T., D.A. and A.T.; Methodology, M.G.T., D.A., M.C. and S.G.; Investigation, A.R.; Resources, A.R.; Data curation, D.A., M.C., A.R. and S.G.; Writing—original draft, M.G.T., D.A. and M.C.; Writing—review & editing, M.G.T., D.A. and M.C.; Visu-

alization, D.A., M.C. and A.T. All authors have read and agreed to the published version of the manuscript.

**Funding:** This article has been supported by the funds of the COST Action CA18110 "Underground Built Heritage as Catalyser for Community Valorisation". DUS.AD017.178.

**Institutional Review Board Statement:** Ethical review and approval were waived for this study, due to the fact that no images or fragile personal data were collected and used, and thus the research did not require a written consent or ethics approval.

**Informed Consent Statement:** Informed consent was obtained from all subjects involved in the study.

**Data Availability Statement:** Data sharing not applicable.

**Acknowledgments:** This article is based upon work from COST Action "Underground Built Heritage as catalyser for Community Valorisation (Underground4value)", CA18110, supported by COST (European Cooperation in Science and Technology). The authors thank the Bishopric of Constantia and Famagusta and the Cyprus Department of Antiquities for their support in the study of the Church of Ayia Napa and the Armenian community of the Holy Sepulcher—Saint Helena Chapel for granting access to the documentation. The authors also thank T. Lerle for sharing the photogrammetric model of the church of Ayia Napa and A. Economou for allowing the use of the monastery plan.

**Conflicts of Interest:** The authors declare no conflict of interest.

## Appendix A

*Appendix A.1 How to Approach Graffiti: A Methodological Overview*

This part aims to give a methodological introduction to the graffiti phenomenon, moving from the authors' experience with medieval and modern graffiti in the Eastern Mediterranean. The first part will describe the workflow developed within the DIGIGRAF (DIGIGRAF: DIGItizing GRAFfiti: methodology definition for the study of Cypriot Historic Graffiti (duration: April 2022–September 2023), an Excellence Hubs project funded by the European Regional Development Fund and the Republic of Cyprus through the Research and Innovation Foundation (Project: EXCELLENCE/0421/0540). It aims to establish a relevant advancement in the state-of-the-art for graffiti study by providing a defined and tested methodology of reference at local and international levels. Moreover, DIGIGRAF will define best practices for the documentation and study of Cypriot graffiti to be adopted by the local authorities and stakeholders for the preservation and promotion of the island's graffiti heritage) project to approach graffiti documentation, analysis and interpretation. The second part will focus on two peculiar aspects:

- The graffiti-making process as a form of visual communication embracing textual and pictorial material and
- The relationship between graffiti and their spatial location as markers of human interaction with the landscape.

*Appendix A.2 Graffiti Workflow*

Within the DIGIGRAF project, a specific workflow has been defined to facilitate the approach to graffiti and offer a clear description of the steps needed for its documentation, analysis, and interpretation.

The workflow is divided into three operational phases: documentation, analysis, and interpretation. The first two consider the graffiti and their physical space—building or site—separately to ensure accurate documentation and analysis, which will flow into the final interpretation step, according to the graffiti/space association mechanism presented in the example above.

The documentation phase aims to document every graffito through the best fitting technique based on the specific characteristics of the material (scratched, dotted, or painted/traced) and its state of preservation. The same is true for the anthropic (building) or natural (site) space, documented by digital technologies to offer a spatial location

to every graffito and to allow an efficient visualization of the space as one of the constitutive elements. Specific guidelines concerning the best fitting techniques based on the physical characteristics of the graffiti and the support/site are being developed to ensure high-quality, reliable, and transparent documentation.

The second step focuses on the description and cataloging of graffiti and its support as a whole through standardized and structured analytical models based on the FAIR Data Principles. The creation of findable, accessible, interoperable, and reusable data has been considered a necessary step to guarantee the sustainability of the process in all its aspects, from production to sharing and reuse [33].

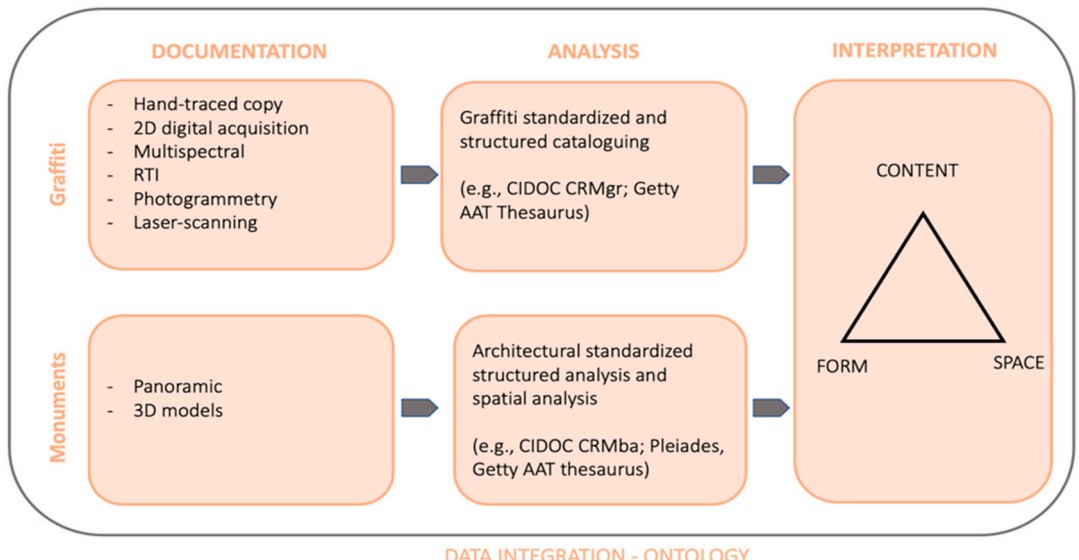

**Figure A1.** Graffiti workflow as developed within DIGIGRAF project ©.

The last step merges the information recorded in the previous steps, interpreting the data in light of the mutual relationships between the three constitutive elements, as presented in the following part.

A final step was the creation of a specific ontology [34] to keep track of all the different phases of work while defining, in a semantic way, the relationships between the different steps and elements involved in the workflow.

This structure was crucial to developing a specific graffiti database included in DIOP-TRA (https://dioptra.cyi.ac.cy/, accessed on 23 July 2023), the Digital Library of the Cyprus Institute-STARC. The DIGIGRAF database will be released by the end of 2023, and it will be organized on two levels, starting from the monument/site to the detection and cataloging of every single graffito. The specific module in DIOPTRA, together with the guidelines for graffiti documentation and cataloging, will offer, on the one hand, valid and innovative tools and solutions for the documentation and cataloging of graffiti and, on the other, a platform for the dissemination and sharing of the Cypriot graffiti heritage. Moreover, the DIGIGRAF platform can be tested in the future with graffiti material from other countries to start creating a shared and inclusive digital tool to connect graffiti scholars and foster data sharing within the academic community and the broader public.

*Appendix A.3 The Graffiti Making Process: How Form, Content, and Space Combine*

The first challenge when working with graffiti is that it is a phenomenon that accompanies human evolution through time and space, responding to different communication and expression needs through different forms—Textual or pictorial [1,35]. Such an extended and varied phenomenon has been approached in many different ways, based on specific historical periods (Prehistory, Antiquity, Middle Ages, etc.), disciplines (Archaeology, Paleography, Anthropology, etc.) and scholars' interests (epigraphy, nautical archaeology, art

history, etc.). The methodological approach presented below focuses on the authors' area of expertise: the Eastern Mediterranean during the medieval and modern periods. Specifically, the paper will present the approaches developed in Cyprus in the frame of DIGIGRAF project and in Israel within the collaboration of the Israeli Antiquities Authorities (IAA) with Hadassah Academic College (HAC). The DIGIGRAF project is developing a specific methodology for the study of Medieval and Modern graffiti, considering different contexts, such as religious and secular buildings, urban and rural areas, underground and above-ground sites. The goal is to consider graffiti in an inclusive way, shaping structured and standardized analytical tools within an ontological frame to ensure their implementation in a wide variety of contexts, and possibly in different time frames. The project's overall objective is to establish good practices for documenting, analyzing, and studying historic graffiti in Cyprus and the Eastern Mediterranean by applying new and innovative digital technologies.

The first step was identifying graffiti's nature and constitutive elements, focusing on its creation process [36]. As mentioned, graffiti uses surfaces not intended for writing, recording a layer of interaction between humans and their surrounding space, natural or anthropic. In archaeology, the creation of graffiti can be defined as an interface: an invisible layer recording an action. In this way, it is possible to underline the existing temporal gap between the creation of the support—or its origin in the case of natural spaces—And the creation of the graffiti. The support is not created to host graffiti, and graffiti cannot exist per se as an object, but just as a result of an intervention on a surface. This aspect is crucial to highlighting the nature of graffiti, which is the result of an action on an existing object rather than the creation of a written artifact—As epigraphs [34]. Their nature encompasses the tangible evidence of a graphic action performed on an existing surface to record intangible practices. This aspect will be better explained in the second part of the paper, during the discussion of the two case studies selected.

After that, the focus shifted to defining the mechanism of creation underlying the making of graffiti. As graphic evidence and a form of communication, semiotics studies helped identify the three constitutive elements involved in the process of graffiti making, defined as form, content, and space [36]. Moreover, a specific ontology was created to define the relationship between the three elements, explaining the graffiti-making process and the reading/deciphering activities [34]. The theoretical model developed represents the key point for the graffiti analysis and study. It defines how the three constitutive elements identified combine in a different way for every graffito to deliver the intended message, offering at the same time an original insight into the people and society that produced it.

*Appendix A.4 How Form, Content and Space Combine in Graffiti Analysis and Interpretation: An Example*

To better explain how the process of making graffiti works, let us see an example from Cyprus, considering one of the most common contents scratched on the walls: identity. Expressing identity through graffiti is very popular. It has been associated primarily with the appropriation of space [37] (pp. 168–170) [38] (pp. 157–162) [39], achieved by marking it with texts or drawings representing the writers' identity, performing the exact mechanism of contemporary tagging [2] (pp. 37–38). The action of 'tagging', in addition to claiming the appropriation of space, also indicates the will to record and fix the passage and presence of an individual in a specific place. Graffiti expressing this function is defined as commemorative, mainly in the context of historic graffiti [40] (pp. 30–38).

The examples have been expressly chosen because they are all located in the same type of context—Christian religious buildings. This choice was made to reduce the variables and make more explicit how minor variations within the three constitutive elements (form, content, and space) define a different function; therefore, they record a different kind of interaction within the same context.

Identity (content) can be expressed in graffiti in both textual and pictorial forms [41] (pp. 293–295). In the first case, the maker will express the content through linguistic codifica-

tion, creating a text with the name and maybe other elements (date, origin, profession, etc.) to reinforce the content. In our case, the writer recorded his name, his provenience, and his profession (Βασίλυοσ μοωαχόσ Μοσχοβόρρώσσοσ 1735—Basil, the Russian-Muscovite monk 1735) [41] (pp. 284–285). In the case of pictorial graffiti, the maker will encode the message through pictorial codification based on visual elements. In this case, the identity can be expressed in various ways, for instance, with a self-portrait or a coat of arms, as in the selected examples (Figure A2).

The three graffiti record the same content—identity—using different forms.

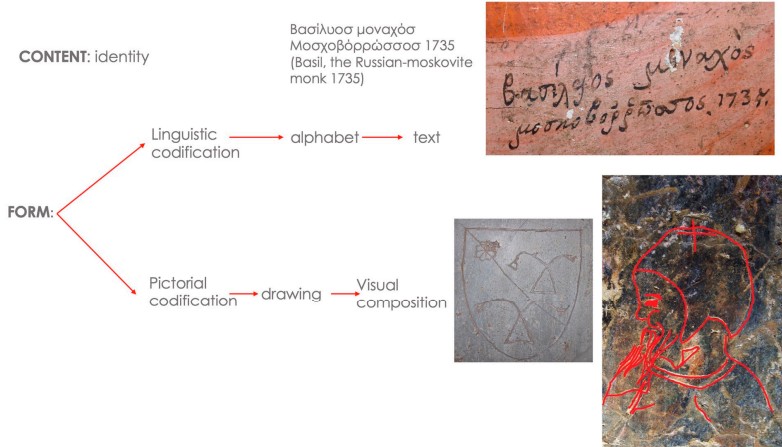

**Figure A2.** Textual and visual codification processes in graffiti (M. Trentin).

The last element to consider is the space, specifically the location of the graffiti within the religious building. The textual graffito is located in a rural, isolated chapel (Saint Andreas Chapel, nearby Kykkos monastery—Troodos mountains), on the iconostasis, just beneath the icon of Christ, on the right side of the Royal Door. The second graffito, the coat of arms, is located on the marble revetment of the central door of the Selimyie mosque—the former Saint Sophia Cathedral—in Nicosia. The third graffito, the portrait, is traced on the south wall of a small rural chapel, known as the Royal Chapel in Pyrga, nearby the south door [41,42] (pp. 286, 294) (Figure A3).

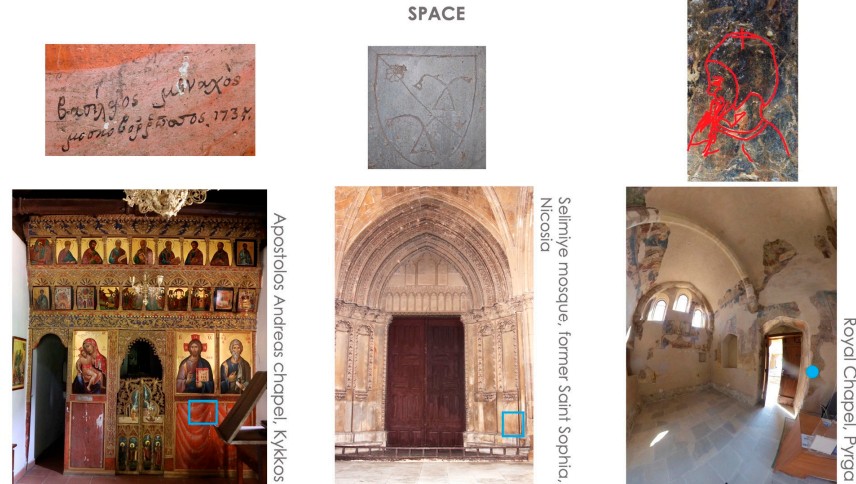

**Figure A3.** Location of the three selected graffiti (M. Trentin).

*Appendix A.5 Spatial Analysis and Graffiti Distribution*

At this point, to understand the function and message conveyed by the graffiti, we need to consider the functions and socioreligious value of the different parts of a Christian

religious building. The parts where the celebration of the liturgy takes place, such as the temple/altar, sometimes delimited by the iconostasis, represent the main focus of the building from a functional and sacred point of view. Other parts have a secondary function and a less relevant level of sacredness, such as the nave, where worshippers participate in the ceremonies, or the narthex [43]. To visualize the different levels of sacredness of the building, within the DIGIGRAF project, a new visual rendering has been developed based on the concept of an influence map.

The sacred-influence map theory revolves around the visualization and analysis of the varying levels of sacred importance within the spaces of a church. It recognizes that different areas within a church hold different degrees of significance based on their liturgical and cult functions. The technique aims to represent these differences in a visually informative and compelling manner. This visualization utilizes a map-based approach, employing a minimalistic design style with concentric circles of different sizes. Each concentric circle represents a specific zone within the church, gradually expanding outward from the most sacred area. The size of the circles is determined based on the level of sacredness attributed to each zone. The larger the circle, the greater the sacred importance of that particular area (Figure A4).

By creating these concentric circles, the Sacred Influence Map offers a visual representation that allows observers to identify and comprehend the hierarchy of sacred spaces within the church. This theory and its accompanying visualization enable researchers, designers, and viewers to better understand the distribution and significance of sacred spaces within a church. It offers a unique perspective on the interaction between worshippers, their acts of devotion, and the areas that hold the highest reverence. The sacred-influence map approach explores the interplay between religious practices, architectural design, and the collective spiritual experience within sacred spaces. It provides valuable insights into the dynamics of devotion and the spatial hierarchy within churches, fostering a deeper appreciation and understanding of these spaces' sociocultural and spiritual significance.

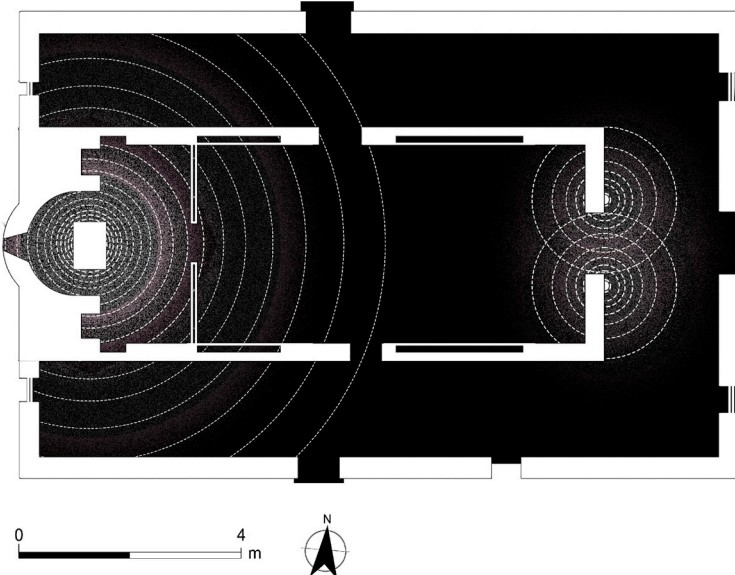

**Figure A4.** Example of a sacred-influenced map for Panagia Podithou (Galata, Cyprus). (Graphics: A. Tinazzo).

Another approach has been implemented to visualize graffiti distribution across the space by adapting the use of heatmaps originally utilized in architecture. This adaptation aims to provide a more comprehensive understanding of the position and characteristics of graffiti left by visitors and pilgrims in churches. By applying a similar concept to architectural heatmaps, it is possible to enhance the identification and visualization of graffiti through the use of color-coded representations (Figure A5). Furthermore, in the field

of graffiti visualization maps, heatmaps can be an invaluable tool for gaining comprehensive insights into various aspects of graffiti. By utilizing graphical representations with colors or intensity levels, heatmaps effectively simplify complex information, making it visually accessible and easily understandable. Heatmaps provide a clear visual representation of graffiti distribution within specific locations or environments. By utilizing varying colors or intensity levels, heatmaps highlight areas with high concentrations of graffiti, enabling researchers to identify popular spots or zones where graffiti is prevalent.

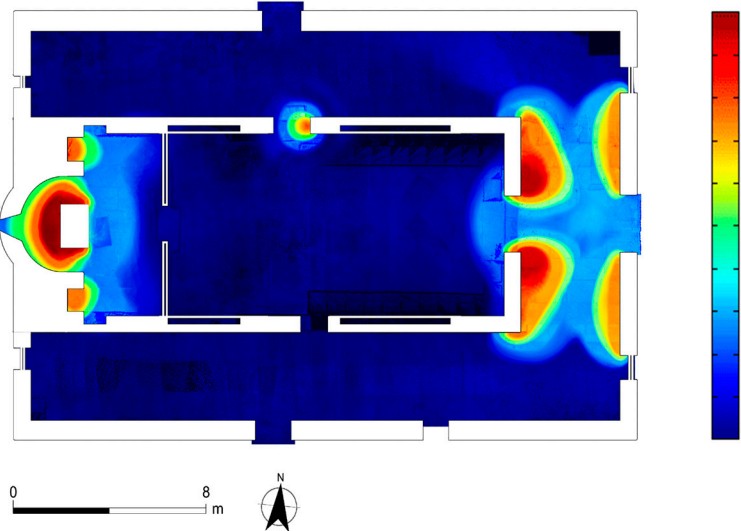

**Figure A5.** Example of a heat map of graffiti distribution for Panagia Podithou (Galata, Cyprus). (Graphics: A. Tinazzo).

In this way, by visually comparing the two maps, the user can gain a first glimpse into the characteristics of graffiti making on the site by comparing the relevance of the different areas with the graffiti distribution (Figure A6).

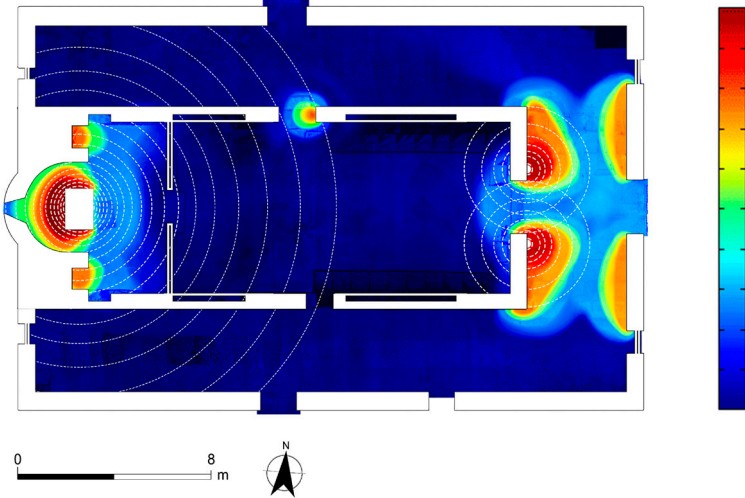

**Figure A6.** Combined visualization of the sacred-influenced map and the heat map on graffiti distribution for Panagia Podithou (Galata, Cyprus). (Graphics: A. Tinazzo).

This visualization system has been developed targeting the general public, aiming to make clearer and more accessible the strong connection between graffiti and their space/context. One of the future steps is to develop a scientific data visualization starting from the graffiti mapping in 3D GIS (Innovative solutions concerning digital applications and methods for spatial documentation and analysis of Rock Art have been developed

within the projects 3D Pitoti (https://3d-pitoti.eu/en/, accessed on 23 July 2023) and Indiana Mas [44], following the example applied to archaeological contexts [45] (pp. 83–95).

Getting back to the example and considering the three graffiti locations in light of the above (Figure A3), we notice that the first graffiti, the monk's inscription, is placed in a visible and very relevant sacred place, on the iconostasis close to the icon of Christ. The second graffito, the coat of arms, is located on the main portal but on the exterior of the building, so in a visible but marginally sacred space. The third and last graffito, the portrait, is located in the nave, an intermediate space inside the building but not so close to the altar.

From there, it is now possible to 'read' the function of the three different identity marks scratched and traced on the walls of the three buildings. In the first case, the identity content acquires religious value due to its prominent location, adding a devotional function to the commemorative one. Placing a name or a mark in a relevant sacred place is fixing the presence of a person in that specific place, establishing a tangible and constant connection between the individual and the sacred space, and ensuring constant divine protection. In the second case, the coat of arms is scratched on the outer part of the building—outside the 'sacred space', in a visible position—The main central door of the building. The primary function is to commemorate the maker's passage from the most relevant urban building of the time: the Cathedral of Saint Sophia. The practice of inscribing coats of arms on religious landmarks was widespread among medieval nobility during their pilgrimages to European and Eastern Mediterranean shrines [29]. This practice was accepted and very rarely recorded in medieval accounts, such as the one of Jacques Le Saige (1518), a silk merchant traveling from Douai to the Holy Land in the first decades of the 16th century, who tells us about the coat of arms and inscriptions visible on the portal of Saint Sophia in Nicosia [41] (pp. 277–278).

The third graffito, the portrait, is located in the nave, inside the religious building, but not in a prominent position. This graffito represents something in between the two previous ones. It aims to commemorate the writer's passage, who placed it inside the building, adding a devotional value to the prevailing commemorative function.

These three graffiti pieces show how the same content—identity—can be expressed in different forms and placed in different spots—in this case, within a Christian religious building—acquiring different values and expressing different functions. The example aimed to present how variations within the three constitutive elements of graffiti—Form, content, and space—Work and must be approached to extract all the potential information.

This structured system of analysis keeps track of all the constitutive elements and relationships involved in the process of visual communication implemented in graffiti. In this way, it is possible to recover all the information, from the single messages, meaning, and function of single graffiti to the more general perception and interaction of past communities with their anthropic and natural landscapes, optimizing the potential of this unique source to explore, from an unconventional and bottom-up perspective, our past.

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
