# Peer review of "Historic Graffiti as a Visual Medium for the Sustainable Development of the Underground Built Heritage"

_sustainability, doi:10.3390/su151511697_

Round 1
Reviewer 1 Report
Please read the comment.

Please read the comment.
Reviewer 2 Report
The article approaches graffiti as a tangible and intangible expression of the interaction between society, culture and landscape, namely adding cultural value (and relevance) to the built underground heritage. The reflection is based on two case studies – in Cyprus and Jerusalem – and aims to demonstrate how graffiti is an element to create connections between landscapes, places and times (past and present). In carrying out the argument of study and reflection, a framework is made about the graffiti and the case studies. The study also involves the use of information and communication technologies. The article is innovative and makes it possible to underline the importance of a given cultural manifestation – graffiti – to revalue the built underground heritage and community.
However, from our point of view, some aspects could be improved, namely:
• There should be at least one image of each of the wider insertion contexts of both case studies. Well, the article refers to the notion of landscape and uses the neologism of grafittiscape, but the reader does not get to understand what the landscape that is being talked about actually is. Namely, bearing in mind that in both case studies the present landscape is quite altered.
• The images that represent a “heat map of graffiti” could have a caption that explains the meaning of the colours, as well as a note that better explains the use made of this technique and its interest.
• The article defends that the graffiti analyzed in both case studies can be considered a “sustainable form of visual communication”. This perspective is defended by the type of surfaces used for the development of graffiti, the fact that this visual expression is inclusive, of be a repository of collective memory and a potential catalyst for current community engagement and a sense of place. However, admitting that such considerations allow the authors to present their arguments in defence of the potential of historical graffiti for the community and the revaluation of the built underground heritage, on the one hand, it does not seem to us that such elements were clearly identified in the presentation of the two case studies. On the other hand, the notion of sustainability deserves a better explanation. What do the authors understand by sustainability? Mainly when the concept of sustainability, at the very least, implies a tripartite interaction between environment, society and economy. What do you mean by “a sustainable form of visual expression”? How is this verified and implemented? For example, how is sustainability promoted and graffiti used as a catalyst for community engagement in the case of Ayia Napa Monastery, namely when there is tourist pressure in its surroundings, effective destruction of the present landscape? How does the idea of a sustainable articulation between past and present actually materialize?
The article is thought to be very interesting and involves in-depth study. Being of great value and importance the arguments defended to underline graffiti's relevance to potentiate and value the underground heritage. But it is not very clear: how this cultural manifestation is sustainable, how it is/can be a catalyst for the engagement of the present community and, assuming that it is a catalyst for local communities, how this can be verified without threatening (putting in risk) the conservation (and respective sustainability) of the underground heritage in which graffiti manifests itself?
Reviewer 3 Report
SUMMARY
The goal of the research paper is to emphasize the significance of graffiti in the study and comprehension of underground built heritage (UBH). The paper presents two case studies, namely the cave church in the monastery of Ayia Napa (Cyprus) and the Saint Helena chapel in the Holy Sepulcher in Jerusalem. By employing digital technologies and a specific workflow, the paper aims to demonstrate how graffiti can be considered as an important element for establishing connections and reinforcing the values of these sites. The research suggests that graffiti, as a sustainable form of visual communication, serves as tangible evidence of intangible socio-cultural contents and practices, complementing traditional historical and archaeological evidence.
COMMENTS
The relevance of the paper to the journal stems from the recognition of historic graffiti as a sustainable form of visual expression. However, there is room for improvement in terms of presentation, specifically in clearly stating the objectives of the two case studies. While the paper's title suggests a focus on detecting human interactions, the case studies primarily delve into the documentation and interpretation of graffiti.
To enhance the paper, I recommend the following revisions to the authors:
1. Clearly articulate the research questions to better outline the study's objectives and provide a more organized approach.
2. Explore existing approaches and solutions proposed for the digital restoration and protection of historic graffiti, also known as petroglyphs. Consider discussing projects such as 3D Pitoti and IndianaMas, which have already addressed the spatial relationships between symbols to derive meaning from graffiti.
3. As a future direction, suggest potential IT solutions for organizing and disseminating this valuable information to the world. For example, propose the establishment of digital libraries or similar platforms to ensure accessibility and preservation of these historical records.
By incorporating these suggestions, the paper can be strengthened, providing a clearer focus on the research questions, acknowledging existing projects, and presenting future prospects for organizing and sharing valuable information related to historic graffiti.
The quality of English is satisfactory.
